

# A comprehensive design schedule for electrosprayed thin films with different surface morphologies

Susan W. Karuga[1], Erik M. Kelder[2], Michael J. Gatari[1,3], Jan C. M. Marijnissen[3]

[1]Department of Electrical and Information Engineering, University of Nairobi, Nairobi, P.O Box 30197-00100, Kenya.

[2]Department of Radiation Science and Technology, Delft University of Technology, Delft, P.O. Box 15, 2629 BJ Mekelweg, The Netherlands.

[3]Institute of Nuclear Science and Technology, University of Nairobi, Nairobi, P.O Box 30197-00100, Kenya.

*Correspondence to*: Susan W. Karuga (swkaruga@uonbi.ac.ke)

**Abstract.** Electrospraying is a technique where a liquid jet breaks up into droplets under the influence of electrical
forces. The technique is outstanding because of its high deposition efficiency and ability to achieve thin films with different surface morphologies. Nowadays, it is applied in the deposition of thin films for nanoelectronics in Li-ion batteries, fuel cells and solar cells, where performance of the deposited layers is determined by their morphologies. Though important in the design of thin films, a systematic way of depositing thin films with desired surface morphologies for optimal operation is not available. In this study, a literature survey has been conducted from which key electrospray parameters have been identified and a
comprehensive design schedule for thin films with different surface morphologies has been developed. To verify the developed schedule, different thin films have been deposited on aluminium foil substrates using lithium salt precursor solutions by altering key electrospray parameters. Surface morphologies of the thin films have been characterized using scanning electron microscopy. Results show three distinct surface morphologies which include porous with agglomerates, porous reticular and dense particulate and they agree with the predictions of the developed design schedule.

**1 Introduction**

Electrospraying is a technique in which a liquid jet is broken up into droplets in the presence of electrical forces. This study focuses on electrospraying in the cone-jet mode and therefore it is discussed in depth. The cone-jet mode is usually obtained when a precursor liquid is pumped through a nozzle at a low flow rate such that dripping is observed in the absence of an electric field. However, when an electric field is applied between the nozzle and a counter electrode and the electric field is
increased stepwise, other modes can be achieved as shown in Fig. 1. These modes include, intermittent cone-jet, spindle (though not shown in Fig. 1), cone-jet and multiple jet modes (Cloupeau and Prunet-Foch, 1994; Swarbrick et al., 2006). Normally, the cone-jet mode is of interest because of its stability and capability to generate monodisperse droplets which are



smaller than the nozzle diameter (Joshi et al., 2021). The precursor solution is pumped through a nozzle and an electrical voltage is applied between the nozzle and a grounded electrode (or vice versa). The resulting electric field creates a charge on

the meniscus of the liquid on the nozzle. Owing to the electric field and the surface charge, the liquid meniscus experiences an electric stress. This electric stress can overcome the surface tension and shape the meniscus into a cone, referred to as the Taylor cone (Taylor, 1964). Charge carriers in the liquid are accelerated towards the cone's apex. They collide with the surrounding liquid molecules causing them to also accelerate. Consequently, a thin liquid jet emerges from the cone's apex which breaks up into highly charged droplets.

Essentially, when depositing thin films by electrospraying, charged droplets are produced from a precursor liquid and they are directed at a substrate of choice. The charged droplets repel one another due to Coulomb repulsive force between them. This results in self-dispersion on the substrate where the solvent is evaporated from the droplet's surface leaving particles to form a solid film. Considering that deposited particle sizes, their monodispersity and distribution on a surface among others defines the quality of the deposited thin film then electrospraying is a powerful deposition technique. This is because it

facilitates the production of uniform films with monodisperse particles in a controlled manner. In their work, Abbas et al. (2017) evaluated the effect of an applied electric field between a nozzle and a ring during spray deposition of $Co_3O_4$ thin films and they referred to this process as electrostatic spray pyrolysis. From their findings, the film produced without an electric field showed defects on the surface like pin holes, cracks and crystal flakes while the film deposited in the presence of an electric field had a smoother more uniform appearance with uniform crystallinity. In another study by Bansal et al. (2012), an

electric field was applied on the substrate and an ultrathin $SnO_2$ film was deposited using the spray pyrolysis technique. The film showed higher stoichiometry, better crystallinity, larger grain size and higher transparency compared to a film deposited without an electric field.

Nowadays, electrospraying is involved in nanotechnology and microelectronics because in such applications the performance of the device is significantly influenced by the film's surface morphology. For instance, in the deposition of

cathode and anode materials for thin films of Li-ion batteries, porous or hollow surface morphologies are preferred because they alleviate capacity fading during cycling by providing enough room for contraction and expansion. Besides, they also offer more reaction sites and provide improved electron transport. On the contrary, the electrolyte layer is required to be dense so that it can be effective in inhibiting short circuits (Bezza et al., 2019; Pei et al., 2016). Electrospraying has also been considered as a breakthrough technique for fuel cell technology because the performance of a fuel cell's catalyst layer depends on its

morphology. Compared to standard electrodes prepared by airbrushing, Conde et al. (2021) reported increased fuel cell performance by ~20 % for porous layers deposited by electrospraying. Also, owing to the homogeneous distribution of the electrosprayed layer over the substrate, Silva et al., (2021) reported an enhanced maximum power of the fuel cell by about 10 %.

In the production of solar cells using the spin coating technique, perovskite materials have made enormous progress

with a power conversion efficiency exceeding 25 %. However, scaling up the production of perovskite solar cells from lab-scale devices to large-scale commercial manufacturing is challenging due to their instability. Also, the conventional methods



used to fabricate the solar cells lead to excess material wastage and they are not compatible with all the layers. Therefore, a scalable method with a high material utilization rate is required for the successful commercialization of perovskite solar cells. Electrospraying offers several advantages for manufacturing perovskite solar cells, including the ability to deposit all the

layers. In addition, electrospray can deposit uniform and high-quality layers for enhanced performance and stability of perovskite solar cells, making them more commercially viable. Several studies have reported superior photovoltaic performance for electrosprayed perovskite solar cells. For instance, in the deposition of perovskite thin films for a solar cell, Chandrasekhar et al. (2016) reported a large variation between films deposited with electrospray and spray pyrolysis in terms of surface morphology, surface coverage and performance. In spray pyrolysis, large droplets were generated leading to slow

evaporation of the solvent. Consequently, a film with voids was formed and also the desired compound material could not be achieved due to incomplete chemical reaction. On the contrary, electrospraying led to formation of desired compound on the substrate due to complete reaction. It also generated smaller droplets that led to a more uniform and dense film and an enhanced efficiency was reported for the electrosprayed film due to efficient electron transfer. While comparing spin-coated and electrosprayed perovskite solar cells, Kavadiya et al. (2017) reported enhanced efficiency and stability for the electrosprayed

device and they attributed this to the achieved uniform and smooth morphology. A power conversion efficiency of 15 % for an all-electrosprayed perovskite solar cell was reported by Jiang et al., 2018 and recently, Wu et al. (2021) reported a power conversion efficiency of 14.4 % for an electrosprayed solar cell compared to 11.1 % for a solar cell produced by doctor blading. The electrosprayed layers were compact and dense ensuring high hole transfer and transport.

It is evident that the surface morphology is critical for enhanced performance in thin films. However, there is limited

understanding regarding the accurate control of thin film morphology and earlier electrospray studies do not give a systematic way of optimizing different parameters to achieve the desired surface morphologies. Therefore, this study identified the parameters that are most relevant for controlling thin film morphology. Using these parameters, a systematic design schedule for electrosprayed thin films with different surface morphologies has been developed. Different experiments have also been performed to verify the schedule.

**1.1 Theory of Electrospray Technique**

In an electrospray experiment charged droplets are generated and directed towards a counter electrode, which can be a selected substrate. Upon evaporation of the solvent in the generated droplets, particles are formed. To estimate the sizes of the droplets and particles formed in cone-jet mode, different authors have derived scaling laws. Electric current flowing through the liquid is a key parameter in the estimation of droplet and particle sizes (Yurteri et al., 2010). In the determination of the jet's electric

current, Ganan-Calvo et al. (1997) presented two distinct profiles, namely flat and non-flat profiles. According to them, liquids with high viscosity and high conductivity have a flat radial profile of the axial liquid velocity in the jet while liquids with low conductivity and low viscosity have a non-flat velocity profile in the jet. To differentiate between these two categories of

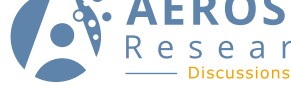

liquids, the same authors developed a dimensionless number (Eq. (1)) later referred to as the viscosity number (VN) by Hartman (1998).

$$VN = \left(\frac{\gamma^3 \varepsilon_o^2}{\mu^3 K^2 Q}\right)^{1/3} \tag{1}$$

where, $\gamma$ is surface tension (N m$^{-1}$), $\varepsilon_0$ is electric permittivity of a vacuum (C$^2$ N$^{-1}$ m$^{-2}$), $\mu$ is liquid absolute viscosity (Pa s), K is liquid conductivity (S m$^{-1}$) and Q is liquid flow rate (m$^3$ s$^{-1}$). For high viscosity and/or high conductivity the VN is relatively low. In practice, a flat radial velocity profile is assumed for viscosity numbers less than or equal to one, while a non-flat radial velocity profile is assumed for viscosity numbers greater than one. For a jet with a flat radial profile, Hartman et al. (1999) derived the equation for electric current as shown in Eq. (2).

$$I^* = b(\gamma K Q)^{0.5} \tag{2}$$

where, I* is jet current for a flat radial profile of axial liquid velocity, b = 2.17 and the other parameters retain the same meaning as in Eq. (1). For liquids with a non-flat radial profile, Hartman (1998) derived a formula to calculate the electric current which was rewritten by Yurteri et al. (2010) in the form of Eq. (3).

$$I = 0.41I^* + \frac{0.24I^{*2}}{E_{z,max}KQ}(Ar_{j0.41}^2 + B) \tag{3}$$

where, I is jet current for a non-flat radial profile of the axial velocity, I* is jet current for a flat radial profile of the axial velocity and $E_{z,max}$, $r_{j0.41}$, A and B are all functions of known parameters. The other parameters retain the same meaning as in Eq. (1). Later, Yurteri et al. (2010) combined these formulas in the form of a ratio as a function of VN leading to Eq. (4).

$$I/I^* = (1 - 0.1 * VN^{0.45})^{-1} \tag{4}$$

Having calculated the jet's electric current, the mechanism by which droplets are formed during jet breakup is also important and must be considered when determining droplet sizes. This mechanism depends on the ratio of the electric normal stress to the surface tension stress on the liquid's surface. A low stress ratio (< 0.3) results in varicose breakup while a high stress ratio results in whipping breakup. In the former, main droplets of similar size are obtained, but in some cases, satellite or secondary droplets may also form, resulting in a bimodal size distribution. On the contrary, whipping breakup leads to a broad size distribution of the main droplets (Yurteri et al., 2010). For both mechanisms, Hartman et al. (2000) derived scaling laws for the main droplet size as shown below.

$$d_{d,varicose} = c_d\left(\frac{\rho \varepsilon_0 Q^4}{I^2}\right)^{1/6} \tag{5}$$

where $d_{d,varicose}$ is droplet diameter in varicose breakup regime, $c_d$ is approximately 2, $\rho$ is liquid density (kg m$^{-3}$), I is jet's electric current and the other parameters retain the same meaning as in Eq. (1). Yurteri et al. (2010) demonstrated that if the radial profile of the axial fluid velocity in the jet is flat, then the current scales according to Eq. (2) and Eq. (5) leads to Eq. (6).

$$d_{d,varicose} = \frac{c_d}{b^{1/3}}\left(\frac{\rho \varepsilon_0 Q^3}{\gamma K}\right)^{1/6} \tag{6}$$

By approximating the values of b and $c_d$ to 2 (as already mentioned above), which gives only a small deviation, Yurteri et al. (2010) obtained Eq. (7).



$$d_{d,varicose} = (\frac{16\rho\varepsilon_0 Q^3}{\gamma K})^{1/6} \tag{7}$$

and in the whipping breakup regime, they obtained the droplet diameter from Eq. (8).

$$d_{d,whipping} = (0.8\frac{288\varepsilon_0\gamma Q^2}{I^2})^{1/3} \tag{8}$$

where $d_{d,whipping}$ is droplet diameter in whipping breakup regime, $\gamma$ is surface tension (N m$^{-1}$), Q is liquid flow rate (m$^3$ s$^{-1}$) and I is jet's electric current. To calculate the droplet size, both Eq. (7) and (8) are used and the smallest value obtained is

assumed to be the correct value. Having determined the size of the main droplets using Eq. (7) or Eq. (8), particle size can be estimated from the main droplet size as shown in Eq. (9) (Yurteri et al., 2010).

$$d_p = \sqrt[3]{f\frac{\rho_{droplet}}{\rho_{particle}}}d_{droplet}^3 \tag{9}$$

where $d_p$ is particle diameter, f is mass fraction of the dissolved material, $\rho_{droplet}$ is liquid density, $\rho_{droplet}$ is final particle density and $d_{droplet}$ is the droplet diameter. However, this is only true for non-porous and non-hollow particles.

135       It is important to note that a stable cone-jet can only be achieved if electrospray is carried out in a limited voltage and flow rate window. The first quantitative description of this stability window was provided by Cloupeau and Prunet-Foch (1989). For selected liquid properties, this window is defined by a minimum flow rate ($Q_{min}$) and a maximum flow rate ($Q_{max}$). The former is the lowest flow rate at which a certain liquid can be sprayed in the cone-jet mode (Hartman, 1998) while the latter is the flow rate beyond which the cone-jet becomes unsteady. Due to the wide variety of complex issues around the

maximum flow rate, there is no formula available to define it (Ganan-Calvo et al., 2018). However, several authors have developed formulas for calculating the minimum flow rate. Among them is Hartman (1998) who suggests that the minimum flow rate is given by Eq. (10).

$$Q_{min} \sim Q_0 = \frac{\varepsilon_o\gamma}{\rho K} \tag{10}$$

Alternatively, Scheideler and Chen (2014) identified different $Q_{min}$ scaling laws for low (Eq. (11)) and high (Eq. (12)) viscous

liquids.

$$Q_{min,low\ viscosity} \sim \frac{\varepsilon\gamma}{K\rho} \tag{11}$$

$$Q_{min,high\ viscosity} \sim \frac{\gamma D^2}{\mu} \tag{12}$$

In these equations $Q_{min}$ is minimum flow rate (m$^3$ s$^{-1}$), $\varepsilon$ is liquid permittivity, $\varepsilon_0$ is electric permittivity of a vacuum (C$^2$ N$^{-1}$ m$^{-2}$), $\gamma$ is liquid surface tension (N m$^{-1}$), K is liquid electrical conductivity (S m$^{-1}$), $\rho$ is liquid density (kg m$^{-3}$), D is outer nozzle

diameter (m) and $\mu$ is liquid viscosity (Pa s). Note that Eq. (11) by Scheideler and Chen (2014) is almost similar to Eq. (10) by Hartman (1998).

      Recently, Marijnissen et al. (2023) extended the work of Hartman and they came up with a formula to calculate $Q_{min}$ if all involved liquid parameters are known.



## 1.2 Key parameters influencing electrosprayed film morphology

From literature, we deduced that the key parameters for designing thin films with different surface morphologies using electrospraying are temperature, flow rate, concentration and deposition time. These parameters are described in detail below but first we consider the study by Rietveld et al. (2006) which provides guidance towards understanding the different parameters.

        According to Rietveld et al. (2006), the morphology of an electrosprayed thin film is determined by the film's growth
rate, the droplet's shear rate and the surface energy interactions between the precursor solution and the substrate. The film's growth rate is defined as the amount of material deposited on the substrate per surface area per second and it is influenced by substrate temperature, flow rate and spray geometry. At relatively high growth rates each deposited droplet spreads on the substrate and before it dries up it encounters other similar droplets leading to coalescence. On the contrary, at relatively low growth rates each deposited droplet dries up independently without coalescing. The droplet's flow on the substrate defines its
shear rate which is obtained from the stress on the droplet at deposition and the droplet's viscosity, that is proportional to concentration. Whether a droplet will spread on the substrate wetting it depends on the surface energy of the substrate and it is characterized by the contact angle of the droplet. Moreover, Rietveld et al. (2006) highlight that commonly used electrospray solutions generate droplets which can spread on the substrate wetting it but they do not discuss the surface energy in details. From Joshi et al. (2021), if the contact angle between the droplet and the substrate is less than 90°, the droplets spread out
wetting the substrate indicating that the surface energy of the substrate is high. Conversely, if the contact angle between the droplet and the substrate is greater than 90°, the droplets show poor wetting on the substrate and tend to form beads indicating that the surface energy of the substrate is low. Considering that the type of substrate in our study is not changing, altering the deposition time is critical in changing the contact angles. For relatively short deposition time the droplets are deposited directly on the surface of the substrate but at a relatively long deposition time, the contact angle changes since droplets are deposited
on top of a layer formed by preceding droplets. Therefore, our deductions on key electrospray parameters are in agreement with Rietveld et al. (2006) and they are discussed in detail below.

        Note that in this study electrospraying was performed using spray geometries that were restricted to short nozzle-substrate distances (2 or 3 cm). According to Rietveld et al. (2006) and Joshi et al. (2021), the spray distance only affects the area covered by the film and the film thickness. In addition, the selected precursor solutions produced droplets that had a
wetting effect on the substrate just like in the study by Rietveld et al. (2006). This meant that the contact angles between the droplets and the substrate were less than 90° as explained by Joshi et al. (2021). Lastly, during droplets evaporation, their sizes reduce but their overall electric charges remain constant. Consequently, Rayleigh limit can be exceeded causing them to explode into many smaller droplets. However, we assumed that the obtained droplet sizes in this study were not affected by this process because most droplets were impinged before Rayleigh break-up just like in the case of Rietveld et al. (2006).




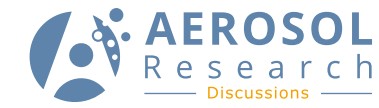

### 1.2.1 Temperature

While keeping other parameters constant different surface morphologies can be achieved depending on the temperature difference between the substrate and the solvent boiling point. This is because, both the solvent boiling point and the substrate temperature play a critical role in droplets' drying. For instance, using a 0.1 M yttria-stabilized zirconia precursor solution in

a solvent mixture of ethanol and butyl carbitol (boiling points of 78 and 231 °C respectively) at a flow rate of 2.8 mL h$^{-1}$ for 5 h, Perednis et al., (2005) reported a wet film at substrate temperature lower than the solvent boiling point (200 °C), a dense film at a substrate temperature of 250 °C (substrate temperature was above solvent boiling point with 19 °C) and a particulate film at a substrate temperature of 300 °C (substrate temperature was above solvent boiling point with 69 °C). A high temperature difference led to evaporation of a big percentage of the solvent from the droplet surface. Therefore, the droplets

were almost dry when they arrived on the substrate leading to formation of discrete particles and a rough film. Conversely, at low temperature but above the solvent boiling point, the droplets were wet when they reached the surface of the substrate leading to spreading and contact between each other to form a smooth film. In their study on the effect of different solvents on particles, Duong et al., (2013) in the preparation of spherical particles used six alcohols and for the different solvents, they obtained different particle morphologies ranging from smooth spherical particles to collapsed shell morphology. They

attributed the difference in particle morphologies to the fact that at a particular substrate temperature, different solvents evaporate at different rates which varies the droplet sizes. Larger droplets resulted in collapsed particles because of the increased mechanical instabilities. Nonetheless, their results could not be fitted in the developed schedule because they did not deposit films but only analysed particles. Lafont et al. (2012) also obtained a more porous film with 1-propanol than with ethanol (boiling points of 97 °C and 78 °C respectively) after electrospraying respective 0.1 M LiNi$_{0.5}$Mn$_{1.5}$O$_4$ precursors at a

flow rate of 1 mL h$^{-1}$ and a substrate temperature of 350 °C (substrate temperature was above solvent boiling point with 253 °C).

### 1.2.2 Flow rate

It has to be noted that among other parameters, flow rate controls the droplet size hence the final particle size. However, flow rate is not an absolute parameter since it is influenced by other factors as shown in Eq. (10) to Eq. (12). Among these factors,

conductivity is the most prominent and its variation can lead to a wide range of droplet sizes. Unfortunately, most of these parameters are not mentioned in the literature cited here. Nonetheless, if the range of conductivity values is not too big it does not tremendously influence the droplet diameter. This is because in the equation for droplet size, the conductivity appears as a power of 1⁄6 or 1⁄3. On the contrary, if the range of conductivities is big the effect on droplet size is significant (Joshi et al., 2013). It is also important to note that flow rates that can achieve the cone-jet mode are defined by a minimum and a maximum

flow rate and they form an operational window. For future research it is recommended that all the involved precursor liquid parameters should be measured and mentioned as indicated in Table 2. At a constant conductivity, low flow rate produces relatively small droplets hence small particles while high flow rate produces relatively big droplets that dry up into big particles.

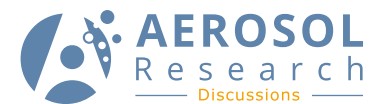

From Kavadiya et al. (2017), electrospraying 14 mg mL$^{-1}$ (0.09 M) $CH_3NH_3PbI_3$ perovskite precursor in isopropyl alcohol (boiling point of 82.5 °C) at flow rates of 0.03, 0.06, 0.09, 0.12 and 0.15 mL h$^{-1}$ at room temperature led to droplets of diameters

505.88, 635.9, 726.94, 799.33 and 860.41 nm respectively and their reported evaporation times were 17.84, 28.22, 36.90, 44.64 and 51.73 μs respectively. The measured particle sizes were 75.36, 77.00, 109.23, 116.31 and 113.43 nm respectively. Therefore, smaller particle sizes were achieved at lower flow rates and they led to the production of smooth uniform films. However, as flow rate increased (above 0.06 mL h$^{-1}$) larger particles were obtained leading to increased film roughness. According to Hong et al. (2017), small droplets have a high rate of solvent evaporation leading to a particulate rough film

while big droplets have a low rate leading to an uneven film with pinholes. Therefore, an intermediate droplet size is required in order to obtain a uniform dense film. This was achieved by electrospraying 30 % wt (2 M) MAPbI$_3$ perovskite liquid precursor in DMSO (boiling point of 189 °C) at a flow rate of 0.05 mL h$^{-1}$ and a substrate temperature of 65 °C for 2 min. The measured droplet size was 4.5 μm which for our case is considered to be large. Different morphological effects based on flow rate were also demonstrated by Ma and Qin (2005) during the electrospray of 0.02 M LiFePO$_4$ precursor solution in a mixed

solvent of ethanol, glycol and butyl carbitol (boiling points of 78 °C, 197 °C and 231 °C respectively) at a substrate temperature of 120 °C (the temperature was lower than the solvent boiling point). At a flow rate of 0.5 mL h$^{-1}$, the generated particles were big (> 400 nm) and they aggregated to form a porous morphology. On the contrary, at a flow rate of 0.05 mL h$^{-1}$ the generated particles were small (< 100 nm) and they formed a uniform dense film. Also, Yu, et al. (2006) reported a porous film with aggregated particles at a flow rate of 4 mL h$^{-1}$ using 0.02 M LiCoO$_2$ precursor in a mixed solvent of ethanol and glycol (boiling

points of 78 °C and 197 °C respectively) deposited for 50 min at a substrate temperature of 350 °C.

### 1.2.3 Concentration

Another factor that influences surface morphology is the precursor liquid concentration. According to Gürbüz et al. (2016), an increase in concentration increases the film thickness which affects morphology. In their study, they electrosprayed SnO$_2$ precursor in ethanol (boiling point of 78 °C) for 1 h at a substrate temperature of 250 °C. Precursor concentrations were varied

from low (0.05 M) to high (0.2 M) but flow rate was kept constant at 7.2 mL h$^{-1}$. A crack free film was obtained from the 0.05 M precursor while a cracked film was obtained after increasing the concentration to 0.2 M. At high concentration, the deposited film was thick leading to a non-uniform drying rate between the top and bottom layers that caused cracking. Also, Bailly et al. (2012) reported a cracked film using a precursor concentration of 0.1 M YSZ in a mixed solvent of ethanol and butyl carbitol (boiling points of 78 °C and 231 °C respectively) at a flow rate of 0.5 mL h$^{-1}$ for 1 h and a substrate temperature of 400 °C. In

another study, Joshi et al. (2013) reported a dense film from a concentration of 0.05 M (which they considered low) $SnCl_4.5H_2O$ precursor in propylene glycol (boiling point of 188.2 °C) at a flow rate of 0.04 mL h$^{-1}$ for 1 h at a substrate temperature of 70 °C.





### 1.2.4 Deposition time

Deposition time is a very important parameter not only in determining the layer thickness but also the surface morphology. In a short deposition time, the film is thin and the droplets get into direct contact with the heated substrate. With increasing time, the film thickens and the substrate surface is completely covered causing consecutive landing droplets to experience varying contact angles that alter the surface morphology. The effects of deposition time on surface morphology were investigated by Gürbüz et al. (2016), who deposited 0.05 M $SnO_2$ film from an ethanol precursor (boiling point of 78 °C) at a flow rate of 7.2 mL $h^{-1}$ and at a substrate temperature of 250 °C for various time intervals. After 20 min, they observed that the substrate was sparsely covered because of the small number of deposited droplets. At 60 min, a lot of droplets had been deposited on the substrate covering the whole surface and leading to a homogenous porous film.  After electrospraying a 0.1 M YSZ precursor in a mixed solvent of ethanol and butyl carbitol (boiling points of 78 °C and 231 °C respectively) at a flow rate of 0.5 mL $h^{-1}$ and a substrate temperature of 400 °C, Neagu et al. (2006) reported a dense coating at 1 h and rough coatings at 4 and 12 h. They attributed the surface roughness to preferential landing of the droplets which occurred at deposition periods longer than 1 h. For Maršálek et al. (2015), they prepared manganese oxide layers from a 0.02 M precursor in a mixed solvent of ethanol and water (boiling points of 78 °C and 100 °C respectively) at a flow rate of 1 mL $h^{-1}$ and a substrate temperature of 200 °C. For their case, deposition times between 10 and 30 min yielded relatively compact and thin layers while longer periods led to a tendency of agglomeration. In the study by Joshi et al. (2015), they obtained porous films using 0.1 M $Bi_2WO_6$ precursor in propylene glycol (boiling point of 188.2 °C) deposited at a flow rate of 0.04 mL $h^{-1}$ for 80 min and a substrate temperature of 120 °C. They reported increased film porosity with deposition time. In another study, Joshi et al. (2012) obtained a dense film using 0.3 M ZnO precursor solutions in propylene glycol (boiling point of 188.2 °C) at a flow rate of 0.075 mL $h^{-1}$ for 30 min and a substrate temperature of 200 °C. At deposition times of 10, 20, 40 and 60 min, which they considered to be short, Yoon et al. (2016) obtained uniform compact films from $WO_3$ precursor in mixed solvent of polyethylene and ethanol (boiling points of 200 °C and 78 °C respectively) at a flow rate of 0.08 mL $h^{-1}$ and a substrate temperature of 80 °C.  In other studies, long deposition time led to a porous reticular morphology. As indicated by Koike and Tatsumi (2005; 2007), droplets spread gradually on the substrate surface and the temperature at the droplet edge was higher than at its centre. Therefore, the solvent at the droplet edge evaporated faster than at its centre. This process led to ring-shaped nucleation and precipitation that formed a reticular morphology characterized by pores and walls. An example is Ma et al. (2014) who electrosprayed 0.1 M MnO precursor in 1,2-dihydroxypropane (boiling point of 188.2 °C) at a flow rate of 1.5 mL $h^{-1}$ for 3 h and a substrate temperature of 240 °C. Another example is J. Yuan et al. (2017) using 2 mM $CoMn_2O_4$ precursor in a mixture of ethanol and 1,2-propanediol (boiling points of 78 °C and 188.2 °C respectively) a flow rate of 2 mL $h^{-1}$ for 4 h and a substrate temperature of 250 °C. Also T. Yuan et al. (2017) using 0.01 M $CoMn_2O_4$ precursor in 1,2-propanediol (boiling point of 188.2 °C) a flow rate of 2 mL $h^{-1}$ for 3 h and at a substrate temperature of 200 °C. The porosity of the film was observed to increase with deposition time as demonstrated by Wang et al. (2011) who used a 0.03 M $V_2O_5$ precursor in a solvent mixture of water, ethanol and 1,2



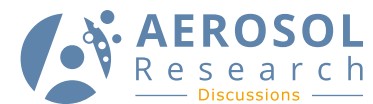

propylene glycol (boiling points of 100 °C, 78 °C and 188.2 °C) at a flow rate of 72 mL h$^{-1}$ for deposition times ranging from 4 to 12 h and a substrate temperature of 260 °C.

## 2 Design schedule

With the aim of enabling the production of thin films with different surface morphologies by electrospraying, a design schedule was developed from literature as discussed in section 1.2. Nonetheless, it was noted that most electrospray studies do not

mention all the parameters but only discuss the ones of interest. For the purpose of this study, only electrospray studies that provided complete information on all the key parameters were selected and cited. Therefore, for future research it is recommended that all the involved parameters should be measured and mentioned. The developed schedule as shown in Table 1 provides a systematic way of obtaining different surface morphologies by just altering key parameters. The different parameters have been defined in terms of being high or low and cut-off values have also been derived from cited literature and

performed experiments. Consequently, all possible combinations of high ($\geq 0.1$ M) or low ($< 0.1$ M) concentration, high (result in big droplets $\geq 1$ µm) or low (result in small droplets $< 1$ µm) flow rate, high (substrate temperature above the solvent boiling point with 52 °C or more) or low (substrate temperature below the solvent boiling point or above the solvent boiling point with less than 52 °C) temperature and long ($> 1$ h) or short ($\leq 1$h) deposition time, are indicated providing a systematic way of designing different surface morphologies.

## 3 Experimental Work


To verify the developed schedule, different electrospray experiments, whose parameters are outlined in Table 3, were carried out using the electrospray set up shown in Fig. 2. It consists of a heated substrate holder, a nozzle (EFD ULTRA), a high voltage power supply (FUG HCN14-12500) connected to the nozzle, a syringe pump (KD Scientific 100) where a precursor solution contained in a syringe is fed in a controlled flow rate via a chemically-resistant hose (Watson-Marlow) to the nozzle

and a temperature controller connected to the substrate holder. The nozzle is held on a movable table which allows the adjustment of nozzle to substrate distance. The lithium nickel manganese oxide (LiNi$_{0.5}$Mn$_{1.5}$O$_4$, LNMO) precursor solutions were prepared by dissolving stoichiometric amounts of reagent grade lithium nitrate (LiNO$_3$), manganese nitrate (Mn(NO$_3$)$_2$·4H$_2$O) and nickel nitrate (Ni(NO$_3$)$_2$·6H$_2$O) in 2-propanol (boiling point of 82.5 °C). The lithium chloride (LiCl) precursor solutions were prepared by dissolving reagent grade lithium chloride (LiCl) in dimethyl sulfoxide (DMSO) (boiling

point of 189 °C). Different precursor solutions were pumped through the syringe at selected flow rates and the nozzle to substrate distance was 2 or 3 cm. For each experiment, the precursor was sprayed through a metallic nozzle of 1.54 mm internal diameter. To create an electric field, the substrate holder was grounded while a high voltage was applied on the metallic nozzle. The voltage applied on the nozzle was adjusted for each experiment to yield a steady cone-jet. At this point the liquid meniscus on the nozzle acquired the shape of a cone that did not relax back to a normal droplet shape. After spraying for a selected



duration, a thin film was deposited on a heated aluminium foil substrate at a selected temperature to evaporate the solvent. The resulting surface morphologies for the films generated from different experiments were analyzed using a scanning electron microscope (JEOL JSM-6010LA).

## 4 Results and Discussion

As outlined in Table 3, distinct thin films were deposited by altering the identified key parameters. Surface
morphologies of the deposited films were then characterized using scanning electron microscopy (SEM). The surface morphologies observed were porous with agglomerates, porous reticular and dense particulate.  The first morphology observed was porous with agglomerates as illustrated in Fig. 3 to Fig. 5. The thin film in Fig. 3 was obtained from a 0.1 M LNMO precursor solution in 2-propanol (boiling point of 82.5 °C) electrosprayed at a flow rate of 1 mL h$^{-1}$ for 3 h and a substrate temperature of 200 °C (substrate temperature was above solvent boiling point with 117.5 °C). The surface morphology was in
agreement with the prediction of the design schedule (Table 1) after electrospraying a high concentration precursor solution at a high flow rate on a substrate that was heated at a high temperature for a long spray duration. A similar morphology was reported by Perednis et al. (2005) who electrosprayed a 0.1 M yttria-stabilized zirconia precursor solution in a solvent mixture of ethanol and 1-methoxy-2-propanol (boiling point of 78 and 120 °C respectively) at a flow rate of 5.6 mL h$^{-1}$ for 5 h and a substrate temperature of 260 °C (substrate temperature was above solvent boiling point with 140 °C).

In Fig. 4, another porous film with agglomerates was observed after electrospraying 0.3 M LNMO precursor in 2-propanol (boiling point of 82.5 °C) at a flow rate of 0.5 mL h$^{-1}$ for 3 h and a substrate temperature of 200 °C (substrate temperature was above solvent boiling point with 117.5 °C). The surface morphology was in agreement with the prediction of the design schedule when a high concentration precursor solution is electrosprayed at a low flow rate on a substrate heated at a high temperature for long spray duration. Neagu et al. (2006) reported a similar morphology after electrospraying a 0.1 M
YSZ precursor solution in a mixed solvent of ethanol and butyl carbitol (boiling point of 78 and 231 °C respectively) at a flow rate of 0.5 mL h$^{-1}$ at 4 and 12 h spray durations and a substrate temperature of 400 °C (substrate temperature was above solvent boiling point with 169 °C). Also, Lafont et al., (2012) obtained a similar morphology after electrospraying 0.1 M LiNi$_{0.5}$Mn$_{1.5}$O$_4$ precursor solution in 1-propanol (boiling points of 97 °C) at a flow rate of 1 mL h$^{-1}$ for 2 h and a substrate temperature of 350 °C (substrate temperature was above solvent boiling point with 253 °C).

As shown in Fig. 5, a porous film with agglomerates was obtained when a 1 M LiCl precursor solution in DMSO (boiling point of 189 °C) was electrosprayed at a flow rate of 0.4 mL h$^{-1}$ for 7 h on a substrate that was heated at 200 °C (substrate temperature was above solvent boiling point with 11 °C). The surface morphology was in agreement with the prediction of the design schedule after electrospraying a high concentration precursor solution at a low flow rate on a substrate that is heated at a low temperature for a long duration. A similar morphology was reported by Joshi et al. (2015) using 0.1 M
Bi$_2$WO$_6$ precursor in propylene glycol (boiling point of 188.2 °C) at a flow rate of 0.04 mL h$^{-1}$ for 80 min and a substrate temperature of 120 °C (substrate temperature was below solvent boiling). They also reported that film porosity increased with





deposition time. The porous with agglomerates morphology on the thin films was attributed to the fact that the generated droplets dried in transit and completely dry particles were deposited on the substrate to form a particulate layer. Subsequently, other particles were deposited on the formed layer and they experience resistance during their discharge on the substrate. As a

result, preferential landing took place in areas where the particles manage to discharge. The particles then adhere on those positions forming agglomerates which were characterized by aggregates or groups of particles as illustrated schematically in Fig. 6.

The second type of morphology was porous reticular as indicated in Fig. 7 to Fig. 9. The thin film in Fig. 7 was obtained when a 1 M LNMO precursor solution in 2-propanol (boiling point of 82.5 °C) was electrosprayed at a flow rate of

2 mL h$^{-1}$ for 3 h and a substrate temperature of 100 °C (substrate temperature was above solvent boiling point with 17.5 °C). The film's surface morphology was in agreement with the prediction of the design schedule after electrospraying a high concentration precursor solution at a high flow rate on a substrate that is heated at a low temperature for long spray duration. A similar morphology was reported by Ma et al. (2014) who electrosprayed 0.1 M MnO precursor in 1,2-dihydroxypropane (boiling point of 188.2 °C) at a flow rate of 1.5 mL h$^{-1}$ for 3 h and a substrate temperature of 240 °C (substrate temperature

was above solvent boiling point with 51.8 °C).

Thin films in Fig. 8 and Fig. 9 were obtained when 0.0375 M LNMO precursor solutions in 2-propanol (boiling point of 82.5 °C) was electrosprayed at a flow rate of 2 mL h$^{-1}$ for 2 h at a substrate temperature of 350 °C and 100 °C respectively (substrate temperature was above solvent boiling point with 267.5 °C and 17.5 °C respectively). The films' surface morphologies were in agreement with the predictions of the design schedule after electrospraying a low concentration precursor

solution at a high flow rate on a substrate that is heated at a high temperature (Fig. 7) or a low temperature (Fig. 8) for long spray duration. T. Yuan et al. (2017) reported a porous reticular film using 0.01 M CoMn$_2$O$_4$ precursor in 1,2-propanediol (boiling point of 188.2 °C) electrosprayed at a flow rate of 2 mL h$^{-1}$ for 3 h and a substrate temperature of 200 °C (substrate temperature was above solvent boiling point with 11.8 °C). Similar morphology was also reported by Wang et al. (2011) who electrosprayed a 0.03 M V$_2$O$_5$ precursor in a solvent mixture of water, ethanol and 1,2 propylene glycol (boiling points of 100

°C, 78 °C and 188.2 °C) at a flow rate of 72 mL h$^{-1}$ and a substrate temperature of 260 °C (substrate temperature was above solvent boiling point with 71.8 °C). Comparing Fig. 7 and Fig. 8, the film became more compact with an increase in substrate temperature. This was also reported by Wang et al., (2009) who obtained porous reticular Fe$_2$O$_3$ films using a 0.005 M precursor in a mixed solvent of 1, 2-propylene glycol and ethanol (boiling points of 188.2 °C and 78 °C respectively) at a flow rate of 2.4 mL h$^{-1}$. At substrate temperatures ranging from 170 to 230 °C, they observed a decrease in pore size with increasing

substrate temperature. The reticular morphology was characterized by pores and walls just like a mesh as illustrated schematically in Fig. 10. This was attributed to the uneven drying of droplets whereby upon arrival on the substrate, they spread gradually with higher temperature at the droplet edge compared to their centre. As a result, the solvent at the droplet edges evaporated faster than at the centre leading to ring-shaped nucleation and precipitation forming a mesh-like morphology.

Lastly, Fig. 11 and 12 show dense particulate surface morphologies. In Fig. 11, the thin film was obtained when a 0.5

M LNMO precursor solution in a solvent mixture of 2-propanol and ethylene glycol (boiling points of 82.5 and 197.3 °C) was

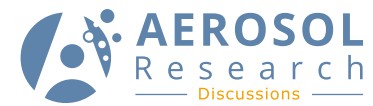

electrosprayed at a flow rate of 2 mL h$^{-1}$ for 1 h at a substrate temperature of 200 °C (substrate temperature was above solvent boiling point with 2.7 °C). The film's surface morphology was in agreement with the prediction of the design schedule after electrospraying a high concentration precursor solution at a high flow rate on a substrate that is heated to a low temperature for a short duration. Joshi et al. (2012) obtained a similar morphology using 0.3 M ZnO precursor solutions in propylene glycol (boiling point of 188.2 °C) at a flow rate of 75 μL h$^{-1}$ for 30 min and a substrate temperature of 200 °C (substrate temperature was above solvent boiling point with 11.8 °C). Also, Hong et al. (2017), reported a dense film after electrospraying 30 % wt (2 M) MAPbI$_3$ perovskite liquid precursor in DMSO (boiling point of 189 °C) at a flow rate of 0.05 mL h$^{-1}$ for 2 min and a substrate temperature of 65 °C (substrate temperature was below solvent boiling point). Perednis et al. (2005) also reported a dense particulate film after electrospraying 0.1 M yttria-stabilized zirconia precursor solution in a solvent mixture of ethanol and butyl carbitol (boiling point of 78 and 230 °C respectively) at a flow rate of 2.8 mL h$^{-1}$ for 1 h and a substrate temperature of 250 °C (substrate temperature was above solvent boiling point with 20 °C).

In Fig. 12, another dense particulate film is shown and it was obtained when a 0.05 M LiCl precursor solution in DMSO (boiling point of 189 °C) was electrosprayed at a flow rate of 0.4 mL h$^{-1}$ for 1 h on an aluminium foil substrate heated at a temperature of 200 °C (substrate temperature was above solvent boiling point with 11 °C). The film's surface morphology was in agreement with the prediction of the design schedule after electrospraying a low concentration precursor solution at a low flow rate on a substrate at a low temperature for a short time. Ma and Qin (2005) electrosprayed 0.02 M LiFePO$_4$ precursor solution in a mixed solvent of ethanol, glycol and butyl carbitol (boiling points of 78, 197.3 and 231 °C) at a substrate temperature of 120 °C (substrate temperature was below solvent boiling point). At a flow rate of 0.05 mL h$^{-1}$ the generated particles were less than 100 nm and they dried to form a uniform dense film. Also, Joshi et al. (2013) reported a dense film from a 0.05 M SnCl$_4$.5H$_2$O precursor in propylene glycol (boiling point of 188.2 °C) at a flow rate of 0.04 mL h$^{-1}$ for 1 h and a substrate temperature of 70 °C (substrate temperature was below solvent boiling point). Similar observations were also made by Kavadiya et al. (2017) who electrosprayed 14 mg mL$^{-1}$ methylammonium lead iodide perovskite precursor solution (0.09 M) in isopropyl alcohol (boiling point of 82.5 °C) at different flow rates ranging from 0.03 to 0.15 mL h$^{-1}$ at room temperature (substrate temperature was below solvent boiling). It was evident that the resulting droplet diameters increased with flow rate and they ranged from 505.88 to 860.41 nm respectively. The droplet evaporation times also increased with droplet sizes and they ranged from 17.84 to 51.73 μs respectively. Upon drying, the resulting particle sizes ranged from 75.36 to 113.43 nm respectively. So smaller particles were achieved at lower flow rates and they led to the production of dense films characterized by small particles that are in close contact as illustrated schematically in Fig. 13.

## 5 Conclusion

Electrospraying is an efficient technique for deposition of thin films with diverse surface morphologies crucial for various applications in micro and nanoelectronics, Li-ion batteries, fuel cells and solar cells. Our study has highlighted the significance of understanding key electrospray parameters in achieving desired surface characteristics. Through literature survey, we

identified these parameters to be temperature, flow rate, concentration and deposition time. In addition, we developed a comprehensive design schedule for thin films with different surface morphologies. The experimental validation of our design

schedule involved depositing thin films on aluminium foil substrates using lithium salt precursor solutions while altering temperature, flow rate, concentration and deposition time. The spray geometry was restricted to short nozzle-substrate distances and the selected precursor solutions generated droplets that exhibited a wetting effect on the substrate. Also, It was assumed that droplets were deposited on the substrate before Rayleigh break-up took place. The resulting surface morphologies, as characterized by scanning electron microscopy, revealed three distinct patterns: porous with agglomerates,

porous reticular and dense particulate. Importantly, these observed morphologies aligned closely with the predictions generated by our design schedule. This research underscores the potential of electrospray deposition as a versatile tool for tailoring thin film properties to meet specific application requirements. Moving forward, further investigations into optimizing electrospray parameters and their effects on thin film morphology could lead to enhanced performance and broader applications across various technological domains.

**Data availability**

All raw data can be provided by the corresponding authors upon request.

**Author contributions**

SWK and JCMM designed the research; SWK performed the experiments; EMK and SWK analyzed the thin films; SWK wrote the manuscript draft; JCMM, EMK and MJG reviewed and edited the manuscript; JCMM, EMK, MJG supervised the

work. All authors made substantial contributions to this work.

**Competing interests**

The authors declare that they have no conflict of interest.

**Acknowledgements**

We acknowledge the International Science Programme at Uppsala University in Sweden for their financial support. The

authors would also like to thank Walter Legerstee for his support during SEM measurements.

**Financial support**

This work was funded by the International Science Programme at Uppsala University in Sweden (Project KEN 01/2, 2017-2021).





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





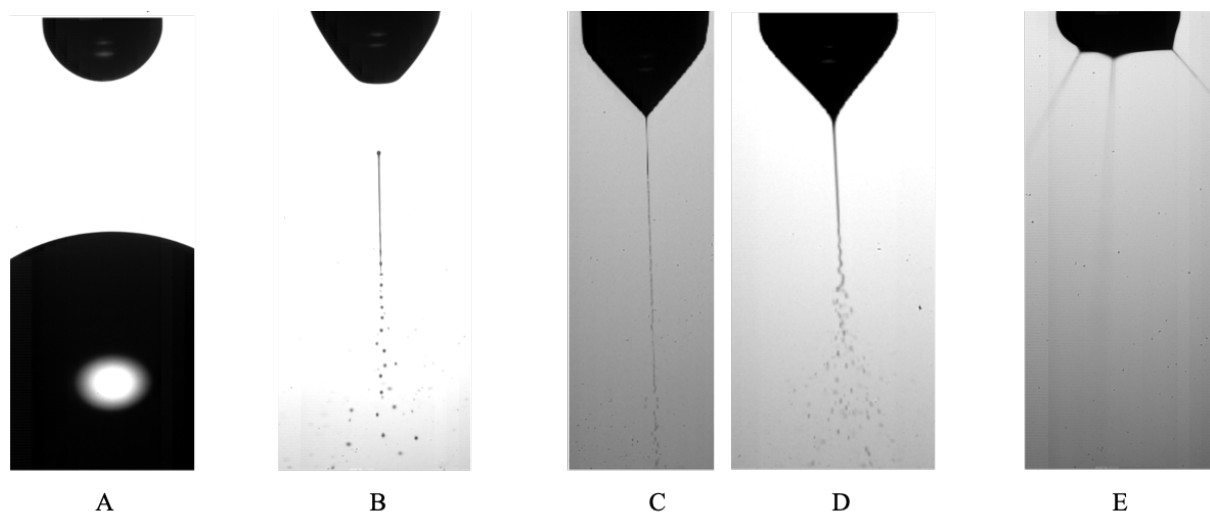

Figure 1: Examples of electrospray modes. A. Dripping, B. Intermittent cone-jet, C. Cone-jet, varicose breakup, D. Cone-jet,
whipping breakup and E. Multiple-jet. Images reproduced with permission from Verdoold et al., 2014 and Yurteri et al., 2010.

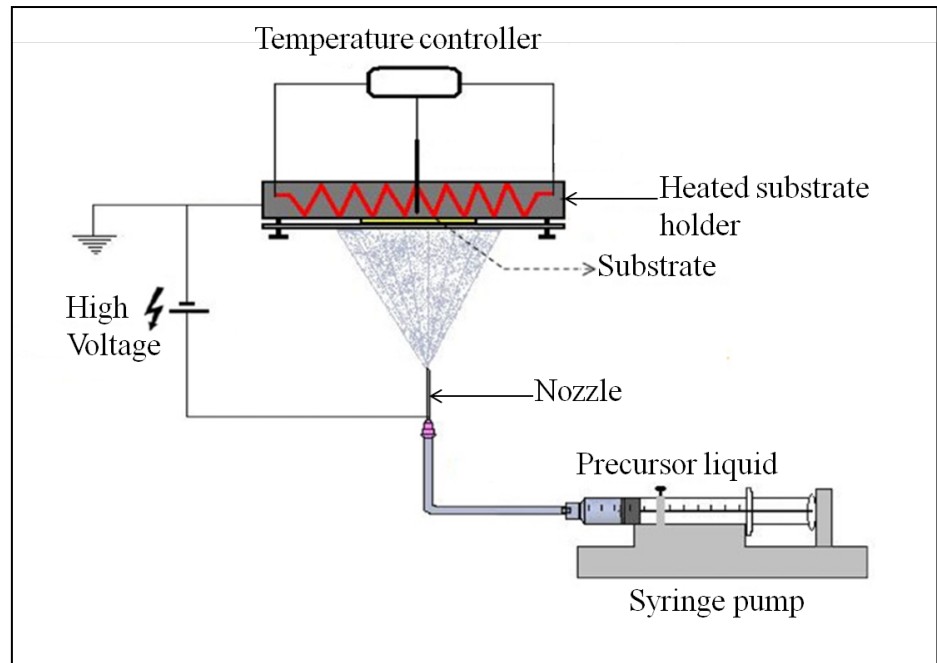

Figure 2: Schematics of the electrospray setup used in this study. Reproduced with permission from Li et al., 2011.




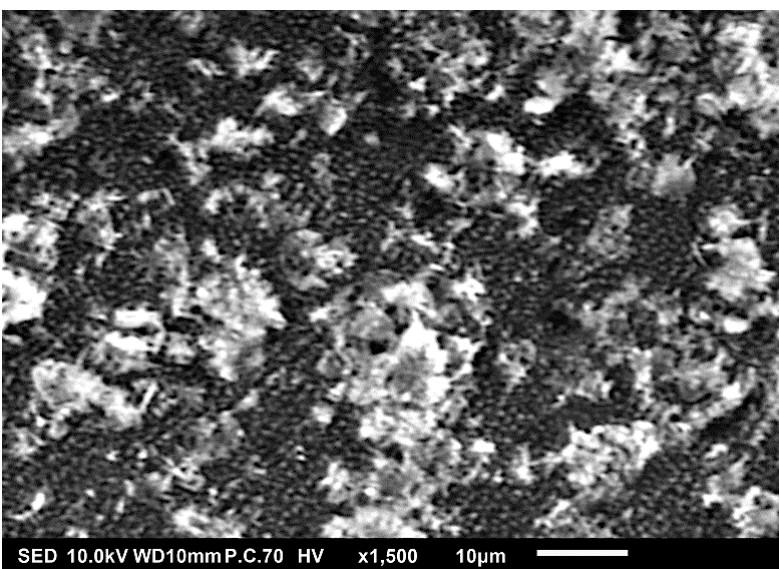

Figure 3: SEM image of a thin film generated by electrospraying 0.1 M LNMO precursor in 2-propanol at a flow rate of 1 mL h$^{-1}$ and a substrate temperature of 200 °C for 3 h.


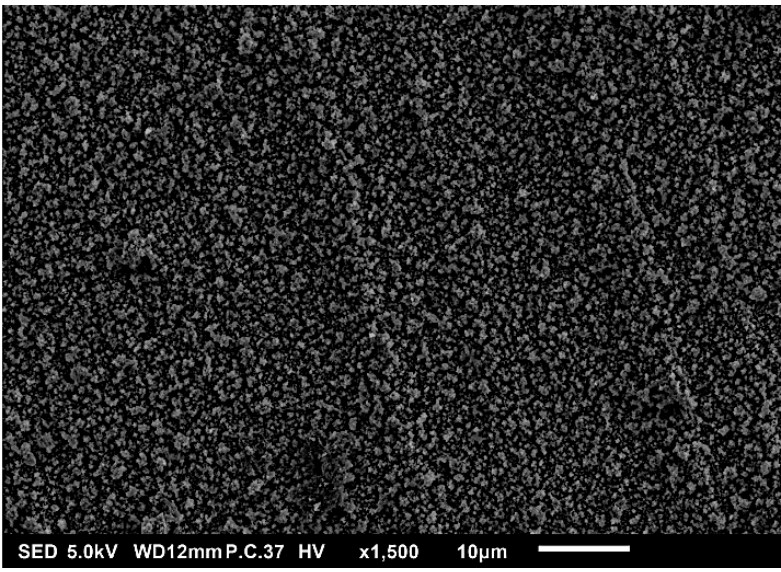

Figure 4: SEM image of thin film generated by electrospray of 0.3 M LNMO precursor in 2-propanol at a flow rate of 0.5 mL h$^{-1}$ and a substrate temperature of 200 °C for 3 h.



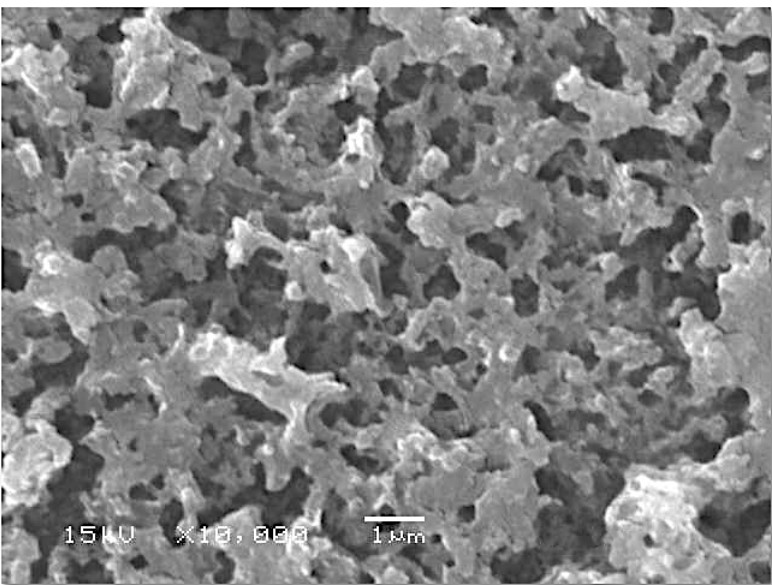

Figure 5: SEM image of thin film generated by electrospray of 1 M LiCl precursor in DMSO (boiling point of 189 °C) at a
flow rate of 0.4 mL h⁻¹ and a substrate temperature of 200 °C for 7 h.

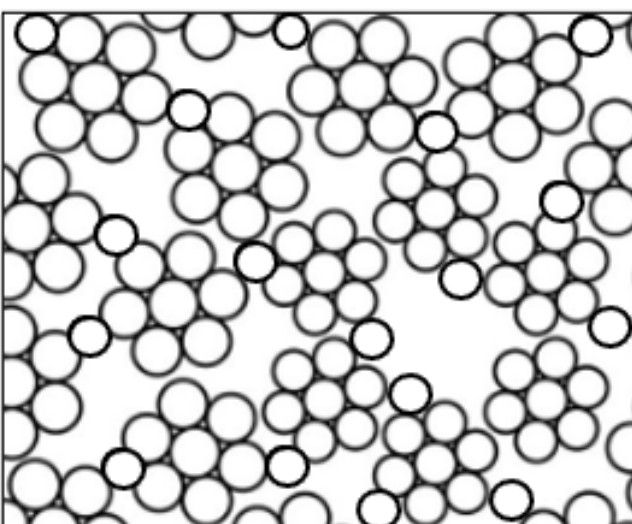

Figure 6: Schematic illustration of a porous film with agglomerates.




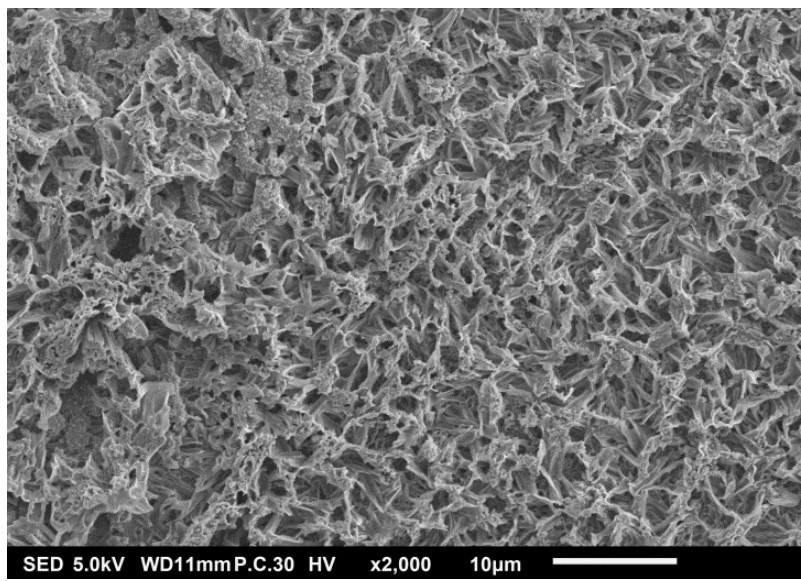

Figure 7: SEM image of a thin film generated by electrospraying 1 M LNMO precursor in 2-propanol at a flow rate of 2 mL h$^{-1}$ and a substrate temperature of 100 °C for 3 h.


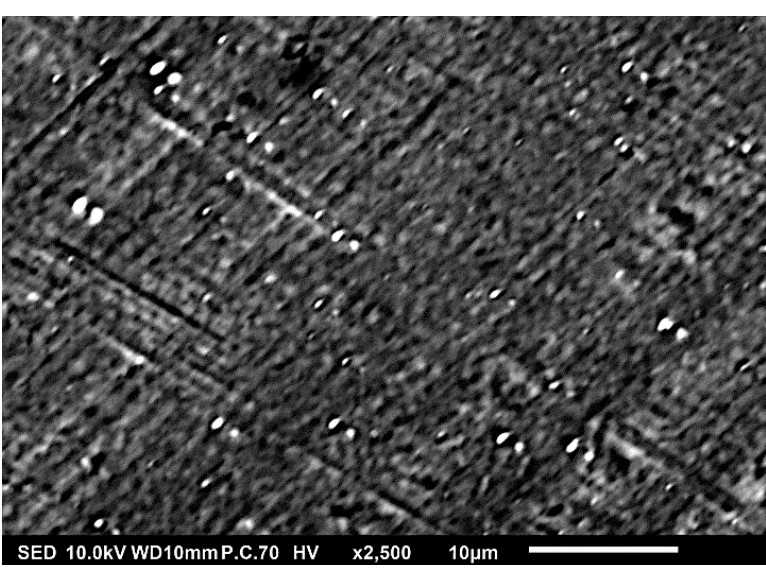

Figure 8: SEM image of a thin film generated by electrospray using 0.0375 M LNMO precursor in 2-propanol at a flow rate
of 2 mL h$^{-1}$ and a substrate temperature of 350 °C for 2 h.



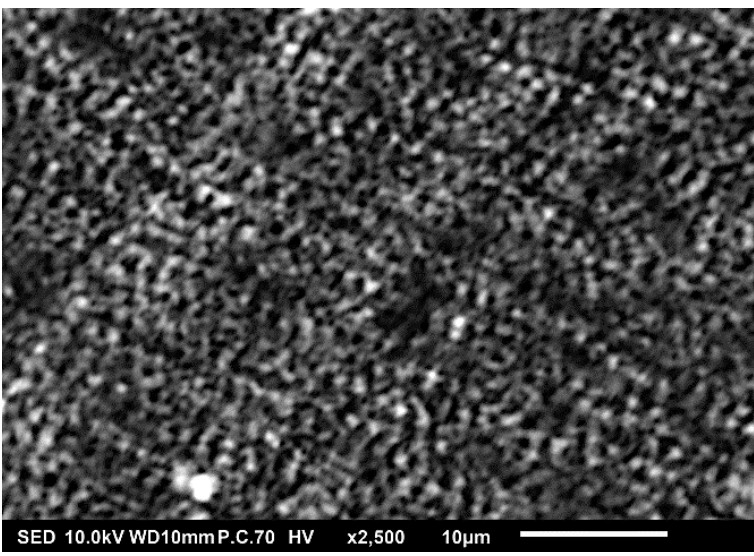

Figure 9: SEM image of a thin film generated by electrospray using 0.0375 M LNMO precursor in 2-propanol at a flow rate of 2 mL h$^{-1}$ and a substrate temperature of 100 °C for 2 h.


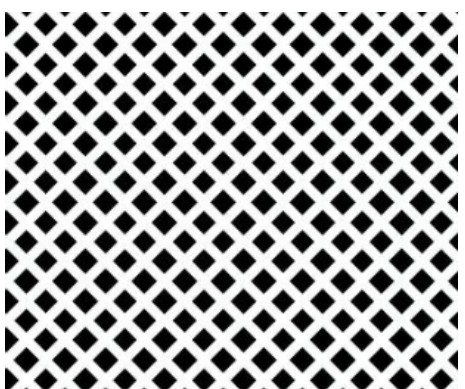

Figure 10: Schematic illustration of a reticular film.



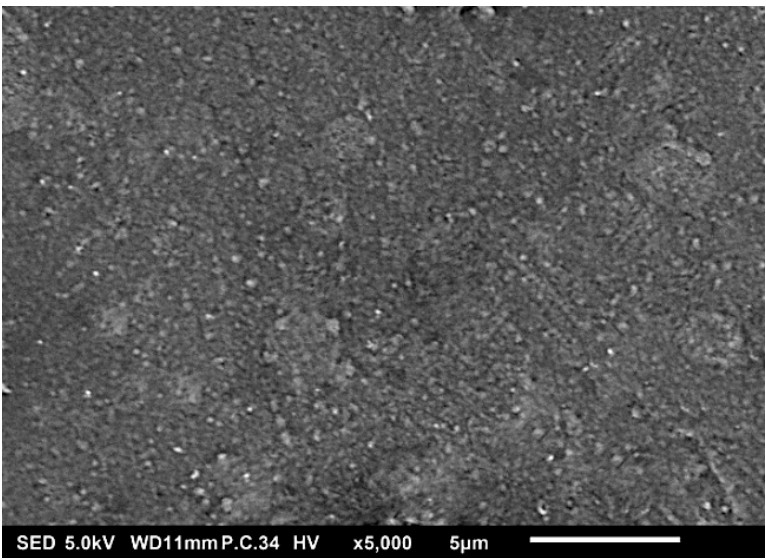

Figure 11: SEM image of thin film generated by electrospray of 0.5 M LNMO precursor in 2-propanol and ethylene glycol at a flow rate of 2 mL h$^{-1}$ and a substrate temperature of 200 °C for 1 h.

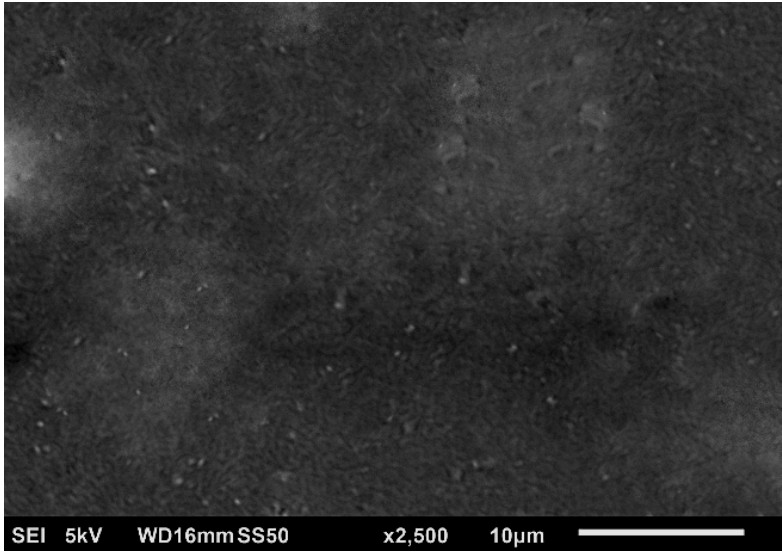

Figure 12: SEM image of thin film generated by electrospray of 0.05 M LiCl precursor in DMSO at a flow rate of 0.4 mL h$^{-1}$

and a substrate temperature of 200 °C for 1 h.



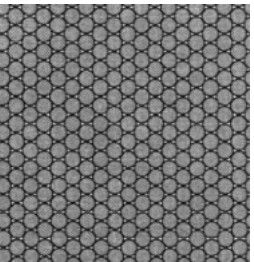

Figure 13: Schematic illustration of a dense particulate film.








Table 1: A design schedule showing how to obtain different surface morphologies by altering experimental and liquid parameters.

| Experimental/precursor liquid parameters | | | | Film morphology | References | Figure |
|---|---|---|---|---|---|---|
| High Concentration*[a] | High Flow rate*[b] | High substrate temperature*[c] | Long deposition time*[e] | Porous: agglomerates | Perednis et al., 2005 | Fig. 3 |
| | | | Short deposition time*[f] | Porous: cracked film | Gürbüz et al., 2016; Bailly et al., 2012 | |
| | | Low substrate temperature*[d] | Long deposition time | Porous: reticular | Ma et al., 2014 | Fig. 7 |
| | | | Short deposition time | Dense: particulate | Perednis et al., 2005; Joshi et al., 2012; Hong et al., 2017 | Fig. 11 |
| | Low Flow rate *[b] | High substrate temperature | Long deposition time | Porous: agglomerates | Neagu et al., 2006; Lafont et al., 2012 | Fig. 4 |
| | | | Short deposition time | Dense: particulate | Neagu et al., 2006; Bailly et al., 2012 | |
| | | Low substrate temperature | Long deposition time | Porous: agglomerates | Joshi et al., 2015 | Fig. 5 |
| | | | Short deposition time | Dense: particulate | Yoon et al., 2016 | |
| Low Concentration | High Flow rate | High substrate temperature | Long deposition time | Porous: reticular | Wang et al., 2011; J. Yuan et al., 2017 | Fig. 8 |
| | | | Short deposition time | Porous: agglomerates | Yu et al., 2006; Gürbüz et al., (2016) | |
| | | Low substrate temperature | Long deposition time | Porous: reticular | T. Yuan et al., 2017; Wang et al., | Fig. 9 |



| | | | | 2011; Wang et al., 2009 | |
| | | Short deposition time | Porous: particulate | Kavadiya et al., 2017; Jo et al., 2014; Ma and Qin, 2005 | |
| Low Flow rate | High substrate temperature | Long deposition time | Porous: agglomerate | Maršálek et al., 2015 | |
| | | Short deposition time | Dense: particulate | Maršálek et al., 2015 | |
| | Low substrate temperature | Long deposition time | Porous: reticular | Koike and Tatsumi, 2005; Koike and Tatsumi, 2007 | |
| | | Short deposition time | Dense: particulate | Kavadiya et al., 2017; Ma and Qin, 2005; Joshi et al., 2013 | Fig.12 |








Table 2: Precursor liquid parameters and estimated droplet sizes.

| No. | Precursor solution | Surface tension (N m⁻¹) | Density (g cm⁻³) | Conductivity (S m⁻¹) | Viscosity (Pa s) | Droplet size (μm) |
|---|---|---|---|---|---|---|
| 1 | 0.1 M LNMO in 2-propanol | 0.02185 | 795 | 0.0595 | 0.001959 | 1.26 |
| 2 | 0.3 M LNMO in 2-propanol | 0.022 | 879 | 0.1501 | 0.002 | 0.74 |
| 3 | 1 M LiCl in DMSO | 0.04346 | 1136.2 | 0.6711 | 0.0051 | 0.42 |
| 4 | 1 M LNMO precursor in 2-propanol | 0.02 | 1119.3 | 0.1775 | 0.0032 | 1.11 |
| 5 | 0.0375 M LNMO in 2-propanol | 0.0223 | 778 | 0.0268 | 0.00238 | 2.08 |
| 6 | 0.5 M LNMO in 2-propanol and ethylene glycol (1:1) | 0.0283 | 946 | 0.1426 | 0.0046 | 1.19 |
| 7 | 0.05 M LiCl in DMSO | 0.04346 | 1010 | 0.1252 | 0.00199 | 0.73 |







Table 3: Parameters for different experiments.

| No. | Precursor solution | nozzle to substrate distance (cm) | Flow rate (mL h⁻¹) | Duration (hours) | Substrate temperature (°C) | Solvent boiling point (°C) |
|---|---|---|---|---|---|---|
| 1 | 0.1 M LNMO in 2-propanol | 3 | 1 | 3 | 200 | 82.5 |
| 2 | 0.3 M LNMO in 2-propanol | 3 | 0.5 | 3 | 200 | 82.5 |
| 3 | 1 M LiCl in DMSO | 2 | 0.4 | 7 | 200 | 189 |
| 4 | 1 M LNMO precursor in 2-propanol | 3 | 2 | 3 | 100 | 82.5 |
| 5 | 0.0375 M LNMO in 2-propanol | 3 | 2 | 2 | 350 | 82.5 |
| 6 | 0.0375 M LNMO in 2-propanol | 3 | 2 | 2 | 100 | 82.5 |
| 7 | 0.5 M LNMO in 2-propanol and ethylene glycol (1:1) | 3 | 2 | 1 | 200 | 82.5 – 197.3 |
| 8 | 0.05 M LiCl in DMSO | 3 | 0.4 | 1 | 200 | 189 |