# Peer review of "A comprehensive design schedule for electrosprayed thin films with different surface morphologies"

_Aerosol Research, 2024_

## Referee Comment (RC2)

This is an interesting paper on a fundamental application of the well-known cone-jet electrospray in materials science: the production of thin films of inorganic materials in critical technological areas such as batteries and fuel or solar cells, to name but a few. The morphologies and mechanical properties of these films are crucial for the performance of the devices. The authors carry out a fairly extensive survey of the literature on the subject, compiling a significant body of accumulated data in order to draw up targeted general "schedules" for obtaining the desired morphologies, based on the operating parameters of the electrospray (essentially the flow rate), which depend on the physical properties of the solutions of the precursors used to form the films.

The paper is well written and organized, and would probably be suitable for publication, but I wonder why the authors chose Aerosol Research rather than a more specific materials' journal, given the approach and ultimate motivation of this paper. In any case, since the generation of an aerosol (more precisely, a fine liquid spray) is a necessary step to form the films objective of this work, it might fit in AR, but in this case I have a number of observations that the authors should address. I will focus on previous literature that I obviously know well:

1- The authors state (line 80) that previous electrospray studies do not provide a systematic way of optimizing different parameters to achieve the desired morphology. After reading the present manuscript, the reader is left with the impression that this work does not represent a drastic improvement on the situation that the authors claim in their statement on the existing literature (a statement with which I do not agree). In fact, while there is very precise evidence for the predictive power of droplet size and charge from well-established scaling laws, the authors use terms such as 'long', 'short', 'high' and 'low' (v.g. Table 1) without any correlation or comparison with precise quantities that could ultimately be reduced to non-dimensional parameters. In this respect, this paper is incremental, but not groundbreaking, because the authors' approach follows the same "cookbook" approach as many other works of this kind. This is not to say that this approach is not valid: on the contrary, it may be the only way to describe a precise know-how in a very complex parametric domain, but in this case the authors should tame the generality of their claims that would justify publication in AR.

2- Please give due credit to previous contributions, and recognize the historic sequence of events in relation to existing scaling laws for the emitted electric current and droplet size:

   a. Expression (2) is the most consistent and accurate scaling law for the emitted electric current in a steady cone jet, but it appears to have been introduced prior to the work of Hartman et al. in 1999: Ganan-Calvo, Barrero and Pantano (1993, J. Aerosol Sci. 24, S19-S20) first presented this scaling law. Please mention this.

   b. Ganan-Calvo (1997, Phys. Rev. Lett. 79, 217-220) introduced for the first time the scaling law (5)-(7) for the droplet size. Please see equation (15) in that work. Please mention this.

   c. If the authors cite Hartman et al. 1999 in relation to (2) and (5)-(7), they must also cite Ganan-Calvo 1999, J. Aerosol Sci. 30, 863-872, who simultaneously introduced the same scaling laws in the same issue of the same journal as Hartman et al. 1999.

   d. Ganan-Calvo et al. (1997, J. Aerosol Sci. 28, 249-275) devoted a section to the minimum flow rate and introduced a criterion similar to (10). Please mention this.

   e. As far as I know, expression (11) was first introduced in Ganan-Calvo et al. 2013, New J. Phys. 15, 033035 (see equation (17) and figure 2 in that work).

   f. In addition, the "minimum" flow rate for viscous fluids given by (12) has already been introduced by Montanero et al. 2011 (Phys. Rev. E 83, 036309, page 5, first paragraph) in the context of the minimum flow rate of flow focusing. Since this phenomenon is closely related to the Taylor cone jet electrospray, since they share a conical meniscus, a short note on the relevance of this expression is needed.

g. Despite the previous point, the same work by Montanero et al. 2011, as well as Ganan-Calvo et al. 2013, discuss the other minima of the flow rate covering the whole parameter space for both Taylor cone jet and flow focusing.

Please mention all this relevant information and provide a general discussion if possible.

3- Lines 165-170: the droplet spreading on the surface depends on two critical parameters, in addition to the viscosity and surface tension: the droplet charge prior to impact, and both the liquid and the substrate conductivity. This also applies to the dried layer left by previous droplets on the substrate.

4- Lines 211-212: Due to its enormous width (many orders of magnitude), the conductivity range of a liquid cannot be described as "not too large" or "not enormously" influencing the droplet diameter in general. This is especially true when we are dealing with solutions exposed to extreme electric fields, whose solutes (e.g. salts) can induce local conductivity values many orders of magnitude different in the same liquid domain, depending on the intensity of the local electric field and local conditions (e.g. a boundary).

5- The authors should take a look at the work of Lopez-Herrera et al. 2023, J. Fluid Mech. 964, A19, and at least mention that electrokinetics is a fundamental aspect to consider in Taylor cone jets when dealing with relatively complex solutions such as inorganic salts in organic liquids, whose dissociation paths can be extremely complex.

6- Line 225: When the authors mention "an intermediate droplet size", ¿how is this intermediate point quantified? ¿How is it quantitatively related to other parameters such as temperature, concentration, substrate, liquid properties and final layer properties? In the same context (and this is related to the previous point 1), ¿is it possible to have a general guideline on the basis of rational relationships between crucial parameters? My guess is that it is indeed possible, with some effort. This would give the paper the relevance that one would expect from the strong claims made both in the abstract and in the conclusions.

7- The authors of the present work make a very important recommendation to the scientific community to disclose ALL related physical properties of the liquid solutions used in the published works, not only those that appear to be of interest to their authors. Again, this should be accompanied by a clear and practical demonstration of the importance of these properties in relation to the properties of the formed layers (the ultimate objective of all this) in quantifiable terms. For example, the authors give a nice guide to the role of the boiling point of the solvent and the excess temperature of the substrate. However, they do not provide any guidance on this excess in relation to the ambient saturation of solvent vapor as a result of the process, ambient pressure, or the thickness of the previous layer already deposited, among several other critical effects.

8- Lines 305-310: One of the crucial parameters to be considered in the electrospraying of complex liquids is the applied polarity. The mobilities of the anions and cations, which give the overall values of the local liquid conductivity, can be radically different, as can the local dissociation reactions, if the applied polarity is positive or negative. Again, please refer to Lopez-Herrera et al 2023 and mention this issue in relation to the present work.

I encourage the authors to carefully consider all of the above points. In summary, this interesting and ultimately valuable paper can be greatly improved by addressing these and other points and suggestions already made by other reviewers.

---

## Author Comment (AC2)

Thank you for your thorough and insightful feedback on our paper. We appreciate the time and effort you have dedicated to providing detailed comments and suggestions for improvement. We are pleased to hear that you found our paper to be interesting and well-written. Regarding your question about the choice of journal, we understand your concern about whether Aerosol Research is the most suitable journal for our paper, given its focus on the generation of aerosols rather than the materials science aspect of our study. We chose Aerosol Research because we believe that our work contributes to the understanding of aerosol generation process, specifically in the context of thin film formation using electrospray technique. The journal of Aerosol Research also provides a unique platform to reach a diverse audience of researchers. We acknowledge your concerns, your expertise and familiarity with previous literature in the field will undoubtedly help us improve the quality and impact of our research. We are committed to addressing the observations you have raised and incorporating any necessary revisions to strengthen our paper as indicated below.

1. We acknowledge your concerns regarding the novelty and significance of our work compared to existing literature. We agree that our work presents incremental progress and it may not be groundbreaking in terms of electrospray methodological innovation. However, we acknowledge that our work provides a systematic way of optimizing different parameters to achieve the desired surface morphologies in the design of thin films. Though important in the design of thin films, a systematic way of depositing thin films with desired surface morphologies for optimal operation has not been provided by earlier studies.

2. In the citation of previous contributions, we apologize for the oversight in not adequately acknowledging previous contributions. We will ensure that proper credit is given to relevant works, particularly those by Ganan-Calvo and others, as per your suggestions.

3. In lines 165-170, we appreciate your input regarding the critical parameters influencing droplet spreading on the substrate. We will revise the relevant section to provide a more comprehensive discussion, taking into account factors such as droplet charge and conductivity.

4. In lines 211-212, we acknowledge the complexity of describing the conductivity range of a liquid and its impact on droplet diameter, especially in the presence of extreme electric fields. We will revise the statement to provide a better understanding on this aspect.

5. Thank you for bringing attention to the work of Lopez-Herrera et al. and the importance of electrokinetics in Taylor cone jets, particularly when dealing with complex solutions. We will incorporate references to relevant literature and discuss the implications of electrokinetics.

6. In line 224, we acknowledge the need for quantifying intermediate droplet sizes and their relationships with other parameters. The statement is made in reference to a study by Hong et al. (2017) who reported that small droplets have a high rate of solvent evaporation leading to a

particulate rough film while big droplets have a low rate leading to an uneven film with pinholes. In order to obtain a uniform dense film, they recommended an intermediate droplet size. From their study, the intermediate droplet size was 4.5 μm and it was achieved by electrospraying 30 % wt MAPbI$_3$ perovskite liquid precursor in DMSO at a flow rate of 0.05 mL h$^{-1}$ and a substrate temperature of 65 °C for 2 min.

7. We appreciate your recognition of the importance of disclosing all relevant physical properties of liquid solutions in published works. We agree that this should be accompanied by a clear and practical demonstration of the importance of these properties in relation to the properties of the formed layers in quantifiable terms. This has been clearly highlighted in section 2 (Design schedule) whereby the terms 'long', 'short', 'high' and 'low' have been defined. We will consider highlighting the same as a note under Table. 1 to enhance clarity and comprehensive understanding.

8. In lines 305 – 310, we acknowledge the significance of considering applied polarity in electrospraying complex liquids. We will reference the work of Lopez-Herrera et al. and mention the implications of applied polarity in relation to our study.

In conclusion, we are grateful for your constructive feedback and assure you that we will carefully address each of your points in the revised manuscript. We believe that incorporating these suggestions will significantly enhance the quality and impact of our paper.

---

## Author Response (AR1)

| Reviewer 1 | | | |
|------|------|------|------|
| No. | Reviewer comments | Response to reviewer comments | Changes in the revised manuscript |
| 1 | The paper provides an overview of the electrospray film deposition parameters that are important for the control of film quality. This is absolutely important and useful for the application of electrospray to generate films with desired properties. The paper is well written and clear to follow and merits publication. I would only suggest to link Table 3 more clearly to the figures that have been produced by the settings listed in Table 3. It also makes cross-referencing between the Tables 1 and 3 easier in my opinion. | Thank you very much for your insightful and encouraging feedback on our paper. We are pleased to hear that you found the overview of electrospray film deposition parameters to be important and useful for practical applications. Your suggestion regarding the clearer linkage between Table 3 and the corresponding figures, as well as facilitating cross-referencing between Tables 1 and 3, is well-taken. We will certainly work on enhancing these aspects to improve the clarity and accessibility of the paper. | In Table 3. a new column labeled "Figure" was inserted and the corresponding figures were added. |
| 2 | I also suggest to add in the abstract and the conclusions that the parameters and the film quality mapping are applicable to inorganic salts, as the results in film quality with organic molecules and in particular polymers are expected to differ more or less extensively depending on their interaction with the electrospray solvent and depending on their crystallisation behaviour, because in particular the amorphous/crystalline behaviour is rather different between polymers and inorganic salts. | Moreover, your recommendation to explicitly mention in the abstract and conclusions that the parameters and film quality mapping are applicable to inorganic salts is valuable. We acknowledge the inherent differences in behaviour between inorganic salts and organic molecules, especially polymers, in response to electrospray conditions and subsequent film formation. Highlighting this distinction will undoubtedly provide greater context and clarity for readers interested in applying our findings across different materials systems. | Lines 15-17: (Abstract) The developed design schedule specifically targets inorganic salts, as the surface morphology of organic salts, particularly polymers, is subject to diverse factors such as solvent interaction and crystallization behaviour.  Lines 312-314: Considering that the surface morphology of organic materials, particularly polymers, is influenced by different |

| | | | factors like their interaction with the solvent and their crystallization behaviour (Rietveld et. al., 2006b), the developed design schedule is only applicable to inorganic salts. |
| | | | Lines 429-431: (Conclusion) The applicability of the developed schedule was restricted to inorganic salts due to the intricate surface morphology and crystallization behaviour exhibited by organic salts. |

| **Reviewer 2** |||||

| No. | Reviewer comments | Response to reviewer comments | Changes in the revised manuscript |
| --- | --- | --- | --- |
| 1 | The authors state (line 80) that previous electrospray studies do not provide a systematic way of optimizing different parameters to achieve the desired morphology. After reading the present manuscript, the reader is left with the impression that this work does not represent a drastic improvement on the situation that the authors claim in their statement on the existing literature (a statement with which I do not agree). In fact, while there is very precise evidence for the predictive power of droplet size and charge from well-established scaling laws, the authors use terms such as 'long', 'short', 'high' and 'low' (e.g. Table 1) without any correlation or comparison with precise quantities that could ultimately be reduced to nondimensional parameters. In this respect, this paper is incremental, but not groundbreaking, because the authors' approach follows the same "cookbook" approach as | We acknowledge your concerns regarding the novelty and significance of our work compared to existing literature.

We agree that our work presents incremental progress and it may not be groundbreaking in terms of electrospray methodological innovation.

However, we acknowledge that our work provides a systematic way of optimizing different parameters to achieve the desired surface morphologies in the design of thin films. | Lines 82-84: the generality of our claims have been tamed and the statement rewritten.

"This work provides a systematic way of optimizing different parameters to achieve the desired surface morphologies in the design of thin films. Therefore, parameters that are most relevant for controlling thin |

| | | | |
|---|---|---|---|
| | many other works of this kind. This is not to say that this approach is not valid: on the contrary, it may be the only way to describe a precise know-how in a very complex parametric domain, but in this case the authors should tame the generality of their claims that would justify publication in AR. | Though important in the design of thin films, such a systematic way of depositing thin films with desired surface morphologies for optimal operation has not been provided by earlier studies. | film morphology have been identified." |
| 2 a) | Expression (2) is the most consistent and accurate scaling law for the emitted electric current in a steady cone jet, but it appears to have been introduced prior to the work of Hartman et al. in 1999: Ganan-Calvo, Barrero and Pantano (1993, J. Aerosol Sci. 24, S19-S20) first presented this scaling law. Please mention this. | In the citation of previous contributions, we apologize for the oversight in not adequately acknowledging previous contributions. We will ensure that proper credit is given to relevant works, particularly those by Ganan-Calvo and others, as per your suggestions in 2(a-g). | Lines 102-103: This scaling law was first introduced by Ganan-Calvo et al. (1993) to show that electric current is a function of liquid properties. |
| 2 b) | Ganan-Calvo (1997, Phys. Rev. Lett. 79, 217-220) introduced for the first time the scaling law (5)-(7) for the droplet size. Please see equation (15) in that work. Please mention this. | | Lines 133-134: It is significant to recognize that Ganan-Calvo (1997) was the first to introduce a general scaling law for estimating droplet size. |
| 2 c) | If the authors cite Hartman et al. 1999 in relation to (2) and (5)-(7), they must also cite Ganan-Calvo 1999, J. Aerosol Sci. 30, 863-872, who simultaneously introduced the same scaling laws in the same issue of the same journal as Hartman et al. 1999. | | Lines 133-135: It is significant to recognize that Ganan-Calvo (1997) was the first to introduce a general scaling law for estimating droplet size. Their study on the jet break up showed that it did not affect the surface charge on the jet (Ganan-Calvo et al., 1999). |
| 2 d) | Ganan-Calvo et al. (1997, J. Aerosol Sci. 28, 249-275) devoted a section to the minimum flow rate and introduced a criterion similar to (10). Please mention this. | | Lines 145-147: However, several authors have developed formulas for calculating the minimum flow rate. Among them are Ganan-Calvo et al. (1997) |

| | | | and Hartman (1998) who suggested that the minimum flow rate is given by Eq. (10). |
|---|---|---|---|
| 2 e) | As far as I know, expression (11) was first introduced in Ganan-Calvo et al. 2013, New J. Phys. 15, 033035 (see equation (17) and figure 2 in that work). | | Lines 155-156: Note that Eq. (11) which was first introduced by Ganan-Calvo et al. (2013) is almost similar to Eq. (10). |
| 2 f) | In addition, the "minimum" flow rate for viscous fluids given by (12) has already been introduced by Montanero et al. 2011 (Phys. Rev. E 83, 036309, page 5, first paragraph) in the context of the minimum flow rate of flow focusing. Since this phenomenon is closely related to the Taylor cone jet electrospray, since they share a conical meniscus, a short note on the relevance of this expression is needed. | | Lines 156-158: Other related studies include Montanero et al. (2011) who pointed out that flow rate is a key parameter in determining jet stability in both low and high viscosity regimes. They introduced the dependence of flow rate on nozzle diameter as indicated in Eq. (12). |
| 2 g) | Despite the previous point, the same work by Montanero et al. 2011, as well as Ganan-Calvo et al. 2013, discuss the other minima of the flow rate covering the whole parameter space for both Taylor cone jet and flow focusing. | | Lines 156-160: Other related studies include Montanero et al. (2011) who pointed out that flow rate is a key parameter in determining jet stability in both low and high viscosity regimes. They introduced the dependence of flow rate on nozzle diameter as indicated in Eq. (12). In their work, $Q_{min}$ increased with viscosity in the low viscosity regime and decreased with viscosity in |

| | | | the high viscosity regime. Also, Ganan-Calvo et al. (2013) studied the forces influencing the stability of liquid ejection in the cone-jet mode and developed different scaling laws for $Q_{min}$ based on viscosity and polarization forces. |
|---|---|---|---|
| 3 | Lines 165-170: the droplet spreading on the surface depends on two critical parameters, in addition to the viscosity and surface tension: the droplet charge prior to impact, and both the liquid and the substrate conductivity. This also applies to the dried layer left by previous droplets on the substrate. | In lines 165-170, we appreciate your input regarding the critical parameters influencing droplet spreading on the substrate. We will revise the relevant section to provide a more comprehensive discussion, taking into account factors such as droplet charge and conductivity. | Lines 186-188: viscosity and surface tension are critical parameters in the determination of the droplet spreading on the substrate. In addition, the droplet charge prior to impact, the liquid and substrate conductivities also play an important role. |
| 4 | Lines 211-212: Due to its enormous width (many orders of magnitude), the conductivity range of a liquid cannot be described as "not too large" or "not enormously" influencing the droplet diameter in general. This is especially true when we are dealing with solutions exposed to extreme electric fields, whose solutes (e.g. salts) can induce local conductivity values many orders of magnitude different in the same liquid domain, depending on the intensity of the local electric field and local conditions (e.g. a boundary). | We acknowledge the complexity of describing the conductivity range of a liquid and its impact on droplet diameter, especially in the presence of extreme electric fields. We will revise the statement to provide a better understanding on this aspect. | For a better understanding on the aspect of conductivity the following statement which was appearing on line 230 was deleted:

"Nonetheless, if the range of conductivity values is not too big it does not tremendously influence the droplet diameter. This is because in the equation for droplet size, the conductivity appears as a power of 1/6 or 1/3. On the contrary, if the range of conductivities is big the effect on droplet size is |

| | | | significant (Joshi et al., 2013)." |
|---|---|---|---|
| 5 | The authors should take a look at the work of Lopez-Herrera et al. 2023, J. Fluid Mech. 964, A19, and at least mention that electrokinetics is a fundamental aspect to consider in Taylor cone jets when dealing with relatively complex solutions such as inorganic salts in organic liquids, whose dissociation paths can be extremely complex. | Thank you for bringing attention to the work of Lopez-Herrera et al. and the importance of electrokinetics in Taylor cone jets, particularly when dealing with complex solutions. We will incorporate reference to relevant literature and discuss the implications of electrokinetics. | Lines 163-167: Though not considered in this work, it might be of interest to study the electrokinetic structure of a steady Taylor cone. This is because the dissociation paths of inorganic salts in organic liquids can be extremely complex leading to formation of either weak electrolyte solutions or strong electrolyte solutions. In the former, ion distribution and conductivity are homogeneous but non-homogeneous for the latter. Therefore, under the same applied voltage, weak electrolyte solutions have larger electrical forces leading to a shorter cone (Lopez-Herrera et al., 2023). |
| 6 | Line 225: When the authors mention "an intermediate droplet size", ¿how is this intermediate point quantified? ¿How is it quantitatively related to other parameters such as temperature, concentration, substrate, liquid properties and final layer properties? In the same context (and this is related to the previous point 1), ¿is it possible to have a general guideline on the basis of rational relationships between crucial parameters? My guess is that it is indeed possible, with some effort. This would give the paper the relevance that one would expect from the strong claims made both in the abstract and in the conclusions. | We acknowledge the need for quantifying intermediate droplet sizes and their relationships with other parameters. The statement is made in reference to a study by Hong et al. (2017) who reported that small droplets have a high rate of solvent evaporation leading to a particulate rough film while big droplets have a low rate leading to an uneven film with pinholes. In order to obtain a uniform dense film, | Lines 240-245: According to Hong et al. (2017), an intermediate droplet size is required in order to obtain a uniform dense film. Though they did not give the limits in droplet sizes, they explained that small droplets have a high rate of solvent evaporation leading to a |

| | | they recommended an intermediate droplet size. From their study, the intermediate droplet size was 4.5 μm and it was achieved by electrospraying 30 % wt MAPbI$_3$ perovskite liquid precursor in DMSO at a flow rate of 0.05 mL h$^{-1}$ and a substrate temperature of 65 °C for 2 min. | particulate rough film while big droplets have a low rate of solvent evaporation leading to an uneven film with pinholes. In their study, the intermediate droplet size was achieved by electrospraying 30 % wt (2 M) MAPbI3 perovskite liquid precursor in DMSO (boiling point of 189 °C) at a flow rate of 0.05 mL h$^{-1}$ and a substrate temperature of 65 °C for 2 min. The measured droplet size was 4.5 μm. |
|---|---|---|---|
| 7 | The authors of the present work make a very important recommendation to the scientific community to disclose ALL related physical properties of the liquid solutions used in the published works, not only those that appear to be of interest to their authors. Again, this should be accompanied by a clear and practical demonstration of the importance of these properties in relation to the properties of the formed layers (the ultimate objective of all this) in quantifiable terms. For example, the authors give a nice guide to the role of the boiling point of the solvent and the excess temperature of the substrate. However, they do not provide any guidance on this excess in relation to the ambient saturation of solvent vapor as a result of the process, ambient pressure, or the thickness of the previous layer already deposited, among several other critical effects. | We appreciate your recognition of the importance of disclosing all relevant physical properties of liquid solutions in published works. We agree that this should be accompanied by a clear and practical demonstration of the importance of these properties in relation to the properties of the formed layers in quantifiable terms. This has been clearly highlighted in section 2 (Design schedule) whereby the terms 'long', 'short', 'high' and 'low' have been defined. We will consider highlighting the same as a note under Table. 1 to enhance clarity and comprehensive understanding. | Lines 671-674: under Table l:

*[1]High concentration is $\geq 0.1$ M while low concentration is $< 0.1$ M. *[2]High flow rate is characterised by big droplets $\geq 1$ µm while low flow is characterised by small droplets $< 1$ µm. *[3]High substrate temperature is a value above the solvent boiling point with 52 °C or more while low substrate temperature is a value below the solvent boiling point or above solvent boiling point with less than 52 °C. *[4]Long deposition time is $> 1$ h while short deposition time is $\leq 1$h. |

| 8 | Lines 305-310: One of the crucial parameters to be considered in the electrospraying of complex liquids is the applied polarity. The mobilities of the anions and cations, which give the overall values of the local liquid conductivity, can be radically different, as can the local dissociation reactions, if the applied polarity is positive or negative. Again, please refer to Lopez-Herrera et al 2023 and mention this issue in relation to the present work. | We acknowledge the significance of considering applied polarity in electrospraying complex liquids. We will reference the work of Lopez-Herrera et al. and mention the implications of applied polarity in relation to our study. | Lines 167-170: Also, when electrospraying with different voltage polarities, the average conductivity for the positive polarity is usually higher than that of the negative polarity. Nonetheless, these differences are negligible for weak electrolyte solutions but significant for strong electrolyte solutions (Lopez-Herrera et al., 2023). In this study, such differences were not expected because only the negative polarity was applied.

Line 327: the substrate holder was grounded while a high voltage of negative polarity was applied. |